# Breeding *Capsicum chinense* Lines with High Levels of Capsaicinoids and Capsinoids in the Fruit

**Siyoung Jang** [1] ⬤, **Minjeong Park** [1], **Do-Gyeong Lee** [1], **Jung-Hyun Lim** [2], **Ji-Won Jung** [2] and **Byoung-Cheorl Kang** [1,*]

[1] Department of Agriculture, Forestry and Bioresources, Research Institute of Agriculture and Life Sciences, Plant Genomics and Breeding Institute, College of Agriculture and Life Sciences, Seoul National University, Seoul 08826, Korea; hoifirstlove@snu.ac.kr (S.J.); minjjs@snu.ac.kr (M.P.); drew21755@snu.ac.kr (D.-G.L.)

[2] Research Institute of Biotechnology, CJ CheilJedang Corp., Suwon 16495, Korea; jh.lim@cj.net (J.-H.L.); jiwon.jung@cj.net (J.-W.J.)

[*] Correspondence: bk54@snu.ac.kr; Tel.: +82-2-880-4563

**Abstract:** Capsaicinoids, which cause a hot sensation when eaten, are uniquely present in pepper (*Capsicum* sp.) and are biosynthesized by combining vanillyl amine with branched fatty acids. A mutation in the gene encoding putative aminotransferase (pAMT)—the enzyme that normally biosynthesizes the capsaicinoid precursor vanillyl amine—leads instead to the biosynthesis of vanillyl alcohol, which combines with branched fatty acids to form capsinoids. Here, we report a method for increasing the capsaicinoid and capsinoid contents using quantitative trait locus (QTL) alleles involved in capsaicinoid biosynthesis in the pericarps of extremely spicy peppers. QTLs for capsinoid contents were detected on chromosome 6 and 10 using an $F_2$ population from 'SNU11–001' and 'Bhut Jolokia (BJ)' ('SJ'). 'SNU11–001' contains high capsinoid contents and 'BJ' contains high capsaicinoid contents in both the placenta and pericarp. These QTLs overlapped QTL regions associated with pungency in the pericarp. 'BJ' was crossed also with 'Habanero' ('HB'), which contains capsaicinoids mainly in the placenta, and the resulting ('HJ') $F_2$ and $F_3$ offspring with 'BJ' genotypes were selected based on QTL markers and the pericarp pungency phenotype. Similarly, $F_2$ and $F_3$ offspring with high capsinoid contents in the pericarp were selected in 'SJ' with reference to 'BJ' genotypes at the QTLs. Through continuous self-pollination, 'SJ' and 'BJ' lines with high capsinoid and capsaicinoid contents, respectively, in both the placenta and pericarp were developed. This study is the first to show that lines containing high levels of capsinoids and capsaicinoids can be bred using pericarp capsaicinoid biosynthesis genes.

**Keywords:** *Capsicum chinense*; pericarp; capsaicinoids; capsinoids; breeding

## 1. Introduction

Peppers (*Capsicum* sp.) contain a unique compound called capsaicin, which confers a spicy taste. Peppers are consumed as raw or cooked fruits or as dried spices and are a major crop, with a production of 38,027 million tons in 2019, from which 4255 million tons of dried fruit was obtained. Almost 60% of peppers are grown in Asia (FAO Statistics, 2019).

Capsaicin and its analogs, such as dihydrocapsaicin and nordihydrocapsaicin, are collectively referred to as capsaicinoids, which are known to be effective in preventing cardiovascular disease and cancer [1]. A substance called capsiate, which has a very similar structure to capsaicin but mild pungency, was first discovered in *C. annuum* 'CH–19 Sweet' [2]. The compounds, including capsiate, dihydrocapsiate, and nordihydrocapsiate, are collectively called capsinoids [3,4]. The major difference between capsaicinoids and capsinoids lies in their molecular structures [4]; capsaicinoids have amide bonds, whereas capsinoids have ester bonds. Capsiate has the advantage of being easy to consume for those who do not enjoy spicy foods, while displaying a medical efficacy similar to that of capsaicin [1].

Capsaicinoids are biosynthesized through two pathways: the phenylpropanoid pathway and the fatty acid pathway [5,6]. Among the genes involved in the capsaicinoid biosynthesis pathway, *Pun1* encodes capsaicin synthase (*CS*), a biosynthetic enzyme acting at the final stage of condensing vanillyl amine with the fatty acids [6,7], and *CaKR1* encodes a putative ketoacyl–ACP reductase that catalyzes the biosynthesis of 8-methyl-6-nonenoic acid in the fatty acid pathway [8]. Recently, the transcription factor CaMYB31 was found to regulate the expression of the capsaicinoid biosynthesis genes including *Pun1*, *Kas*, *BCAT*, *C4H*, *Comt*, and *pAMT* [9–11]. Putative aminotransferase (pAMT) catalyzes the biosynthesis of the capsaicinoid precursor by converting vanillin to vanillyl amine at the end of the phenylpropanoid pathway [12]. A mutation in *pAMT* can cause vanillyl alcohol to be produced instead of vanillyl amine, resulting in the biosynthesis of capsinoids instead of capsaicinoids [12,13]. The capsaicinoid biosynthesis gene *Pun1* is also required for capsinoid biosynthesis [14].

Previous studies have used genome–wide association studies and QTL mapping studies to identify the QTLs that regulate capsaicinoid contents in pepper. Among them, QTLs located on chromosomes 3, 6, and 10 were identified as common QTLs in different populations [15–17]. These QTL regions contain previously characterized capsaicinoid biosynthesis genes, including those encoding pAMT, cinnamate 4–hydroxylase (C4H), 4coumarate CoA ligase (4CL), acyl–ACP thioesterase (FatA), and caffeoyl shikimate esterase (CSE) [16]. Extremely pungent peppers such as *C. chinense* 'Bhut Jolokia (BJ)' and 'Trinidad Scorpion', biosynthesize capsaicinoids not only in the placenta but also in the pericarp [18]. To identify the genetic factors that regulate capsaicinoid biosynthesis in the pericarp, a QTL analysis was performed, leading to the identification of QTLs on chromosomes 4, 6, and 11. *KR* and *Ankyrin* were identified as candidate genes underlying the QTL on chromosome 6 [19].

Several series of studies have revealed that *pAMT* mutant alleles are involved in capsinoid biosynthesis. SNPs, insertions, or deletions in exons have generated premature stop codons and frameshift mutations in *pAMT*, and many *pAMT* variants contain *Tcc* (Transposon of *C. chinense*) in introns [12,13,20]. These mutant alleles have been used for marker-assisted selection (MAS) to increase capsinoid contents in pepper. Jang [21] and Jeong [22] designed markers based on the 'SNU11–001' *pAMT* mutant allele, which has *Tcc* in the 3rd intron. Furthermore, several varieties containing capsinoids, such as 'Maru Salad', 'HC3-6-10-11', and the 'Dieta' series, were developed by MAS of *pAMT* mutant genotypes [23–25]. Four 'Dieta' varieties were bred by crossing highly pungent *C. chinense* 'BJ' and 'Infinity' with 'Aji Dulce'. Both 'BJ' and 'Infinity' are pungent, but 'BJ' is highly pungent. Two varieties, 'Dieta0011–0301' and 'Dieta00110602', from 'BJ' contain higher contents of capsinoids compared to the others from 'Infinity'.

In this study, we carried out experiments to explore whether the capsaicinoid and capsinoid contents in the whole fruit can be increased by introducing the genetic factors that regulate capsaicin biosynthesis in the pericarp.

## 2. Materials and Methods

### 2.1. Pedigree Method for Development of 'SJ' and 'HJ' Lines

Three *C. chinense* accessions, 'SNU11–001', 'HB', and 'BJ', were used to construct two populations, 'HJ' ('HB' × 'BJ') and 'SJ' ('SNU11–001' × 'BJ') (Figure 1). Nine 'HJ' $F_3$ plants were selected, which have the highest capsaicinoid contents in pericarp and 'Jolokia'—type genotypes for three QTLs among 192 $F_3$ individuals (Table S1) [19]. Twenty-nine 'HJ' $F_4$ individuals from nine $F_3$ lines were grown, and 17 plants were selected. Thirty–three 'HJ' $F_5$ individuals were grown, and 11 plants were selected by capsaicinoid contents of fruits and yield (Table S2). Thirty–four $F_6$ individuals from 11 $F_5$ plants were grown and capsaicinoid contents were measured to select six plants (Table S3). Six to eight replicates of six $F_7$ lines were evaluated by capsaicinoid contents and three 'HJ' lines were selected (Figure S1).

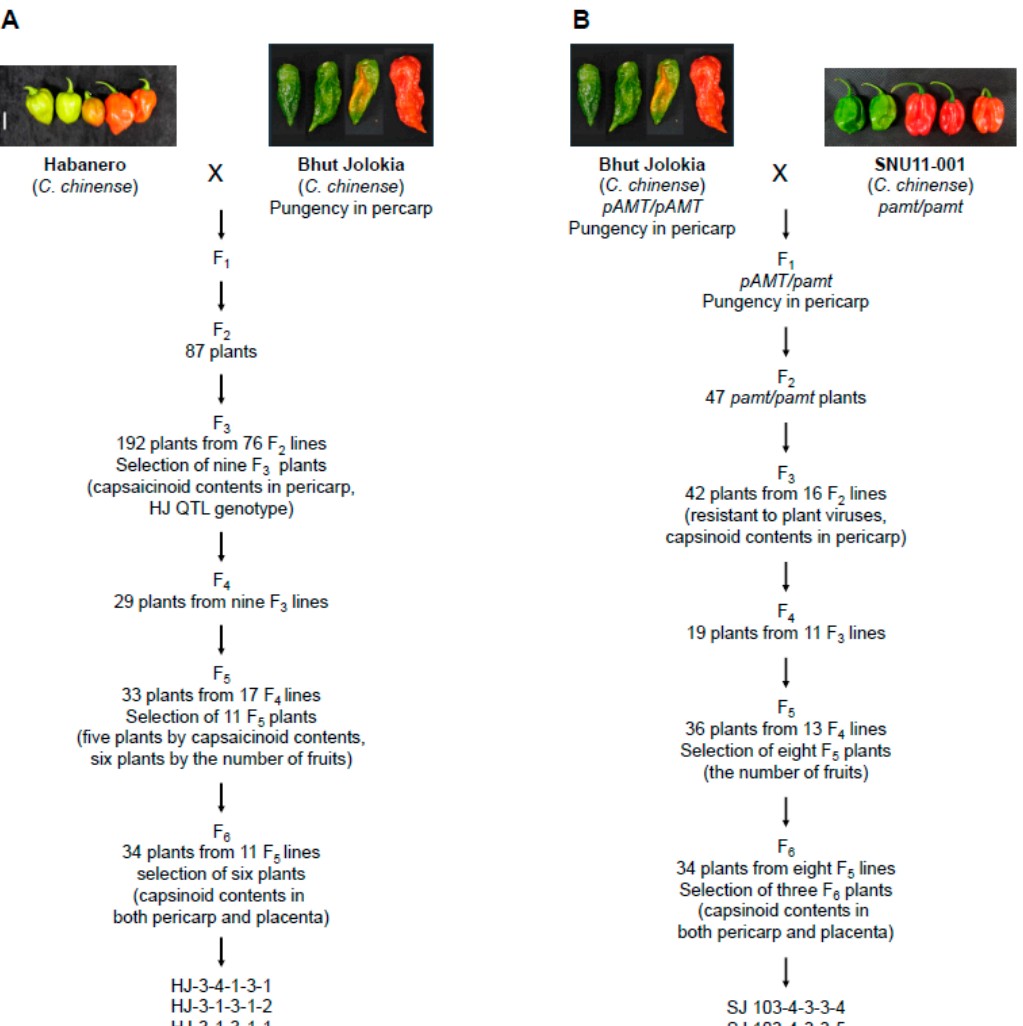

**Figure 1.** Diagram of the novel *Capsicum* variety breeding program. (**A**) Selection of highly pungent lines from the 'HB' × 'BJ' population, 'HJ'. (**B**) Selection of lines containing high levels of capsinoids from the 'SNU11–001' × 'BJ' population, 'SJ'.

Forty–seven 'SJ' $F_2$ plants were selected by *pAMT* genotype analysis among a total of 173 individuals (Figure 1, Table S4). Forty–two 'SJ' 42 $F_3$ plants were grown, however, only 11 individuals were survived due to virus disease, and four plants had the highest capsinoid contents (Table S5). Nineteen 'SJ' $F_4$ individuals from 11 $F_3$ plants were grown. Thirty–six 'SJ' plants from 13 $F_4$ plants were grown, and eight plants were selected by yield and the number of seeds. Thirty–four 'SJ' $F_6$ individuals from eight $F_5$ plants were grown and three $F_6$ plants were selected by capsinoid contents of fruits (Table S6). Similar to the 'HJ' $F_7$ generation, six to eight replicates of three $F_7$ lines were evaluated by capsinoid contents and two 'SJ' lines were selected (Figure S1). The seeds of all breeding lines were stored in Seoul National University.

### 2.2. Growing Region and Season

The 'SJ' $F_3$ lines were grown in an open field at Seoul National University farm, Republic of Korea, in 2017 spring to autumn; the 'SJ' and 'HJ' $F_4$ lines were grown in the greenhouse at Anseong, Republic of Korea, in 2018 spring to autumn; the 'SJ' and 'HJ' $F_5$ lines were grown in Khon Kaen Thailand in 2018 winter to 2019 spring; the 'SJ' and 'HJ' $F_6$ lines were grown in a glasshouse with an average temperature of 25 °C in the green house at Seoul National University farm, in 2019 spring to autumn; the final 'HJ' and 'SJ' varieties were grown at Seoul National University farm in 2020 spring to autumn.

### 2.3. QTL Analysis of the Capsinoid Contents of the 'SJ' $F_2$ Plants

WinQTLcart 2.5 [26] was used for the QTL analysis. Bin map information was generated from 'SJ' $F_2$ genotyping-by-sequencing SNP data [19]. The QTLs were detected for two groups: 172 plants from the entire population and 47 *pamt* mutant plants. Six traits were analyzed: capsiate, dihydrocapsiate, and total capsinoid contents in both the placenta and pericarp. The QTLs were detected using composite interval mapping (CIM). The LOD threshold was determined using 1000 permutations with a 5% probability for each chromosome and each trait.

### 2.4. QTL Genotyping Analysis of Pungency in the Pericarp

The interaction of three QTLs from 'HJ' $F_{2:3}$ was previously shown to explain almost 50% of the variance in their capsaicinoid contents, while another three QTLs from 'SJ' wild–type *pAMT* $F_2$ plants could explain 49% of their capsaicinoid contents' variance [19,27]. The three 'HJ' QTLs are located 223.4 Mbp and 226.5 Mbp along chromosome 6, and 38.8 Mbp along chromosome 11, respectively; these were converted to Cleaved Amplified Polymorphic Sequences (CAPS) (HJ16_CAP6.3_qtlseq_caps, HJ16_CAP6.4_caps and HJ17_CAP11_caps). The three 'SJ' QTLs are located 210 Mbp along chromosome 4, and 231.9 Mbp and 233.5 Mbp along chromosome 6; these were converted into a CAPS marker (SJ_TCP6.1_caps) and two HRM markers (SJ_TCP4.1_hrm and SJ_TCP6.1_hrm). SNP information about the 'HJ' and 'SJ' QTL markers is presented in Supplementary Table S7 and Supplementary Figure S2.

To analyze the HRM markers, the reaction mixture contained 80 ng of DNA, 10 × Hifi buffer, 2.5 mM dNTPs, 10 pmol/μL of each primer, 1 unit of Taq polymerase, and 0.6 μL of Syto9. The PCR conditions were as follows; 95 °C for 5 min, followed by 55 cycles of 94 °C for 20 s, 58 °C for 20 s, and 72 °C for 40 s, with a final extension at 72 °C for 10 min. For the HRM analysis, one cycle was added at the end. The temperature was increased every 2 s from 65 °C to 90 °C to dissociate double–stranded DNA with Syto9 into single–stranded DNA. To analyze the CAPS marker, the reaction mixture was similar to that used for the HRM marker except without the Syto9. For the PCR step, the annealing temperature was 58 °C and the elongation time was 1 min. For the digestion of the PCR product, enzymes *Sau*96I, *Rsa*I, and *Taq*I were used. The PCR products were digested at 37 °C using *Sau*96I and *Rsa*I, and at 65 °C by *Taq*I for 4 h. The digested DNA was loaded on a 1.5% agarose gel.

### 2.5. MAS Using the pAMT Genotype

*pAMT* KASP markers were designed to detect the 8–bp insertion that causes an early stop codon in the 'SNU11–001' *pamt* mutant allele. The *pAMT* KASP marker is composed of three primers with the following sequences: 'Normal_*pAMT*–FAM' 5′–FAM tail–TTGGGAGGCCACAGAAAAAG–3′, 'SNU_*pAMT*–HEX' 5′–HEX tailGCCACACCGCCAC-AGAAAAAG–3′, and '*pAMT*–common reverse' 5–GTAGGTGAAGATGGTGTGGTATTACA–3′. The PCR mixture components and reaction conditions are described in Supplementary Tables S9 and S10.

### 2.6. Analysis of Capsaicinoid and Capsinoid Contents

The placenta and pericarp tissues were dissected out of three fruits from each plant and pooled per plant. After freeze–drying and grinding these tissues, 0.1 g of the pericarp powder and the whole placenta sample were used for the extractions. The capsaicinoids and capsinoids were extracted and analyzed using High Performance Liquid Chromatography (HPLC), as described by Han [14]. The HPLC was performed using Ultimate3000 HPLC (Thermo Dionex, Waltham, MA, USA) at the National Instrumentation Center for Environmental Management (Seoul, Republic of Korea).

One–way ANOVA was used to confirm significant differences in capsaicinoid or capsinoid contents of between breeding lines and parental lines and to evaluate whether capsaicinoid contents increased significantly if three QTLs had more 'BJ'–type genotype in 'SJ' *pAMT* mutant segregation group.

## 3. Results

### 3.1. Capsaicinoid and Capsinoid Contents of Different Accessions

The capsaicinoids and capsinoids were quantified for three *C. chinense* accessions, which are known to have high levels of these compounds. In 'Habanero (HB)' and 'Bhut Jolokia (BJ)', the capsaicinoid content was much higher than that of the capsinoids (Table 1). The capsaicinoid contents of the placenta were 130,520 μg/g of dry weight (DW), 67,741 μg/gDW, and 153 μg/gDW in 'HB', 'BJ', and 'SNU11–001', respectively, while the respective capsaicinoid contents in the pericarp were 3736 μg/gDW, 15,606 μg/gDW, and 16 μg/gDW. The capsaicinoid contents of the 'HB' placenta were about 1.93 times higher than that of 'BJ', while for the pericarp it was 4.18 times higher in 'BJ'. In addition, the average capsaicinoid content per fruit was higher in 'BJ' (15,868 μg/DW) than in 'HB' (13,390 μg/DW). 'SNU11–001' contained almost no capsaicinoid, but had a high capsinoid content in the placenta (21,452 μg/DW).

**Table 1.** Capsaicinoid and capsinoid contents of three accessions.

| Accession | Target Trait | Capsaicinoids (μg/gDW) | | Capsinoids (μg/gDW) | | *pAMT* Genotype |
|---|---|---|---|---|---|---|
| | | Placenta | Pericarp | Placenta | Pericarp | |
| Habanero | Capsaicinoids | 130,520 | 3736 | 4957 | 226 | *pAMT/pAMT* |
| Bhut Jolokia | Capsaicinoids in pericarp | 67,741 | 15,606 | 4142 | 782 | *pAMT/pAMT* |
| SNU11-001 | Capsinoids | 153 | 16 | 21,452 | 1937 | *pamt/pamt* |

### 3.2. QTL Analysis for Capsinoid Contents in 'SJ' $F_2$

A QTL analysis was performed on the capsinoid contents of the pericarp in the $F_2$ 'SJ' population derived from a cross between *C. chinense* SNU–001 and BJ. The QTLs associated with the capsiate, dihydrocapsiate, and total capsinoid contents were commonly found in the lower arm of chromosome 6, and additional dihydrocapsiate QTLs were identified on chromosomes 1 and 10 (Table 2). The QTL region on chromosome 6 was consistent with the QTL associated with capsaicinoid content (Figure 2). These QTLs were analyzed in the subpopulation consisting of the *pamt* mutant plants. The dihydrocapsiate-associated QTL *mSJ–perdhcst10* was found on chromosome 10 and partially overlapped with *SJ–perdhcst10*, which was present in the entire population. QTLs associated with the capsaicinoid and capsinoid contents of the 'SJ' $F_2$ population were present in a common location at 226.4 to 237.5 Mbp along chromosome 6. Markers prepared from the three QTLs showing epistasis, which explained 49% of the capsaicinoid content variation in individuals with the wild–type *pAMT* allele, were therefore used to analyze the *pamt* mutant individuals (Table S4) [27].

**Table 2.** QTLs associated with the capsinoid contents in the pericarp of the entire 'SJ' $F_2$ population and *pamt* mutant plants.

| Segregation Group | Trait | QTL | Chr. | Bin Marker | LOD | $R^2$ (%) | Location (cM) | Location (Mbp) |
|---|---|---|---|---|---|---|---|---|
| Total population | Capsiate | *SJ–percst6.1* | 6 | SJ6_119 | 5.4 | 1.37 | 104–107.5 | 226.4–227.3 |
| | | *SJ–percst6.2* | 6 | SJ6_127 | 6.5 | 5.22 | 113.2–117.4 | 232.5–234.8 |
| | Dihydrocapsiate | *SJ–perdhcst1* | 1 | SJ1_1 | 4 | 9.98 | 0–9 | 16.4–16.9 |
| | | *SJ–perdhcst6* | 6 | SJ6_127 | 4.3 | 2.92 | 107.8–117.4 | 227.8–234.8 |
| | | *SJ–perdhcst10* | 10 | SJ10_108 | 6.8 | 5.87 | 96.8–101.3 | 218.1–223.5 |
| | Total capsinoids | *SJ–pertcps6.1* | 6 | SJ6_119 | 5.1 | 1.04 | 104.1–107.5 | 226.4–227.3 |
| | | *SJ–pertcps6.2* | 6 | SJ6_124 | 6 | 3.41 | 107.5–113.2 | 227.3–232.5 |
| | | *SJ–pertcps6.3* | 6 | SJ6_129 | 6.1 | 3.81 | 117.4–126.9 | 234.8–237.5 |
| *pamt* mutant plants | Dihydrocapsiate | *mSJ–perdhcst10* | 10 | SJ10_bin104 | 7.28 | 20.11 | 89.6–99.3 | 209.9–218.8 |

*PERCST*, capsiate in pericarp; *PERDHCST*, dihydrocapsiate in pericarp; *PERTCPS*, total capsinoids in pericarp; *mSJ*, SJ $F_2$ *pamt* mutant plants group.

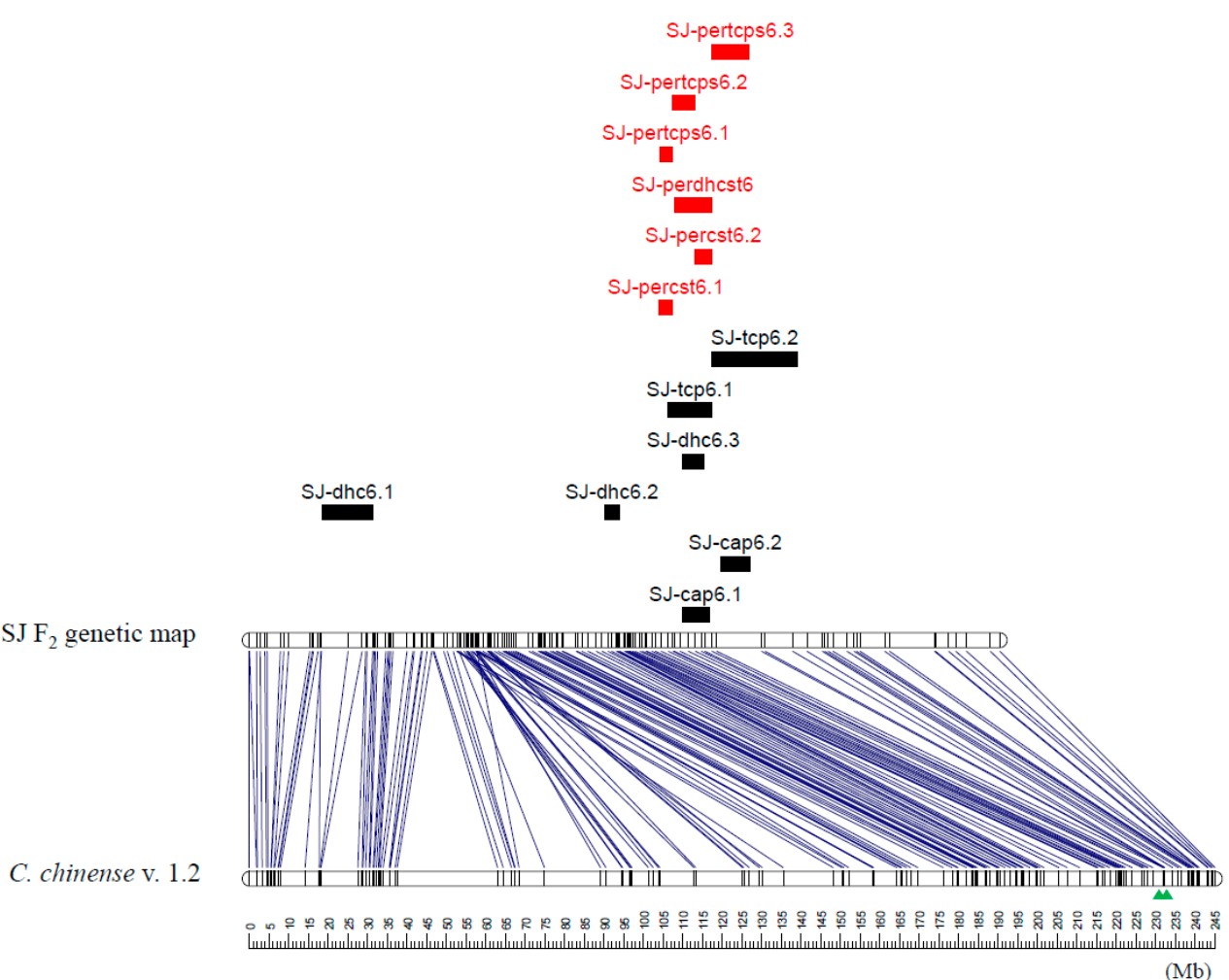

**Figure 2.** 'SJ' F$_2$ linkage map of chromosome 6 containing QTLs related to the capsaicinoid and capsinoid contents in the pericarp. Red and black bars indicate QTLs for capsinoids and capsaicinoids, respectively. Blue lines indicate overlapping bin markers on the genetic map and *C. chinense* scaffold v. 1.2. The two green triangles below the physical map indicate the region of SNP markers based on the QTLs for capsaicinoids.

### 3.3. Primary Selection of 'HJ' Pungent Lines

To develop extremely pungent pepper lines, the 'HJ' population was created by crossing 'HB' and 'BJ' (Figure 1; Figure S1). The distribution of capsaicinoid contents in the F$_2$ population ranged from 282 µg/gDW to 51,563 µg/gDW. Three SNPs linked to 'HJ' QTLs were converted to cleaved amplified polymorphic sequence (CAPS) and high-resolution melting (HRM) markers and used for genotype analysis (Table S7). As a result, we determined that the more 'BJ' genotypes present for the three QTLs the higher the capsaicinoid content (Figure S3).

Through high–performance liquid chromatography (HPLC) and QTL marker analysis, the seven top F$_3$ individuals containing high levels of capsaicinoids were selected, including line 3–3 (40,941 µg/gDW), which had the highest capsaicinoid contents in the pericarp (Figure S3). These top individuals were derived from five F$_2$ individuals, with three of them being derived from the F$_2$ individual with the highest capsaicinoid contents (51,563 µg/gDW).



### 3.4. Capsaicinoid Contents of the Three 'HJ' Selected Lines

Through repeated selection for capsaicinoid contents in pericarp and generation advancement, 11 $F_5$ 'HJ' lines were selected, which originated from six $F_4$ individuals (Figure 1). Assuming that the pericarp DW is proportional to the size of the fruit, the estimated total contents of capsaicinoid for each fruit should take into account their DW; therefore, the $F_6$ and $F_7$ lines were selected for both high capsaicinoid contents and high DW in the placenta and the pericarp. The capsaicinoid contents of the $F_6$ and $F_7$ lines were measured in both placenta and pericarp of the fruit, and finally, the three most pungent lines were selected (Figure 3). The capsaicinoid contents of the 34 $F_6$ plants ranged from 75,039 µg/gDW to 189,298 µg/gDW in the placenta and from 8558 µg/gDW to 67,008 µg/gDW in the pericarp. The ratio of capsaicinoids in the placenta relative to the pericarp ranged from 2 to 16.

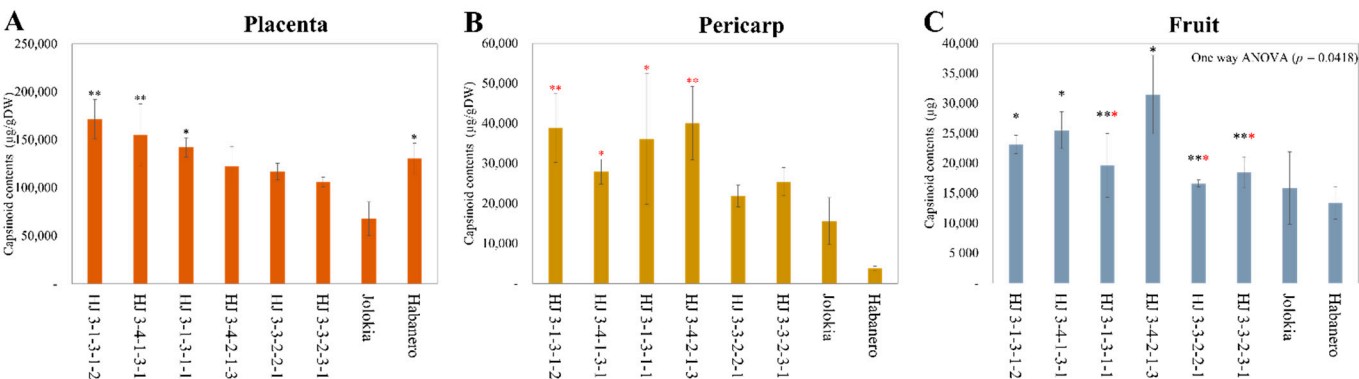

**Figure 3.** Capsaicinoid contents of the placenta (**A**), pericarp (**B**), and fruit (**C**) in the 'HJ' F7 lines. Bars represent standard error (*n* = 3). Black and red asterisks indicate a statistical difference from 'BJ' and 'HB' respectively, as revealed using a one–way ANOVA and Fisher LSD, with *p*–values between 0.001 and 0.01 showed with '**', and lower than 0.05, '*'.

The three $F_6$ individuals with the highest contents of capsaicinoids in one fruit were HJ 3-4-1-3-1 (65,171 µg), HJ 3-4-2-1-3 (44,184 µg), and HJ 3-1-3-1-1 (53,974 µg) (data not shown). The three individuals with the highest contents of capsaicinoids in the placenta were HJ 3-3-2-2-1 (189,298 µg/gDW), HJ 3-3-2-3-1 (172,418 µg/gDW), and HJ 3-1-3-1-2 (179,812 µg/gDW) (data not shown). These six $F_6$ individuals were selected, and their pungency phenotype was analyzed in $F_7$ fixed lines. The capsaicinoid contents in the placenta were significantly different from 'BJ' in three lines, HJ 3-1-3-1-2, HJ 3-4-1-3-1, and HJ 3-1-3-1-1. The capsaicinoid contents of the pericarps of four lines including HJ-3-1-3-1-2 and HJ 3-4-2-1-3 showed a significant difference from 'HB'. All six lines had a significant difference in fruit capsaicinoid content compared with 'BJ', and three lines were different from 'HB' statistically. Of the six lines, three (HJ 3-1-3-1-2, HJ 3-4-1-3-1, and HJ 3-1-3-1-1) were selected for further analysis due to their high average capsaicinoid contents in the placenta, pericarp, and whole fruit.

### 3.5. Primary Selection of 'SJ' Pamt Mutant Lines

To develop lines with high capsinoid contents in the pericarp, 'SNU11–001' (*pamt/pamt*) was crossed with 'BJ' (Figure 1, Figure S1). In the F2 population ('SJ'), the *pAMT* alleles were segregated. To select individuals with high capsinoid contents, 47 individuals homozygous for the *pamt* allele were first identified (Table S8; Figure S4). The capsinoid contents of the *pamt/pamt* individuals ranged from 171 µg/gDW to 11,744 µg/gDW in the pericarp and from 688 µg/gDW to 64,492 µg/gDW in the placenta (Table S4).

To explore whether the pericarp capsinoid contents were higher in plants with more 'BJ' alleles in the capsinoid QTLs, the capsinoid contents of these plants were evaluated and analyzed using an ANOVA (Figure 4). We identified a significant difference (*p*–value

0.0283) between plants with different combinations of 'BJ' alleles demonstrating a positive correlation between the number of 'BJ' alleles and the pericarp capsinoid content.

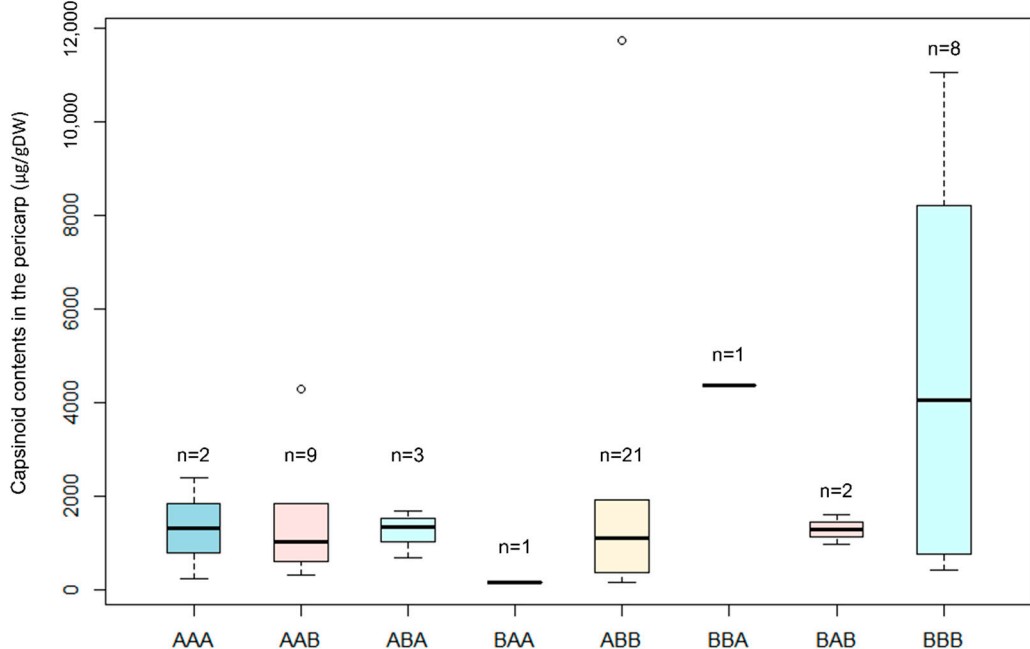

**Figure 4.** Pericarp capsinoid contents of the nine genotypes across three 'SJ' QTLs. The order of the three markers is SJ_TCP4.1_hrm, SJ_TCP6.2_hrm, and SJ_TCP_6.1_caps, with A and B indicating the 'SNU11–001' and 'Jolokia' alleles, respectively. The number of *pamt* mutant $F_2$ plants identified for each genotype is indicated above each column. Empty circles indicate outlier values.

The capsinoid contents in the pericarps of 42 $F_3$ 'SJ' individuals derived from 16 $F_2$ individuals were measured (Figure S5). The capsinoid contents in these individuals ranged from 18 µg/gDW to 4279 µg/gDW (SJ 103–4). Eight individuals with a capsinoid content of 1134 µg/gDW or higher were selected. In addition, three additional $F_3$ plants were selected from the offspring of the $F_2$ line SJ 151, which had a high capsinoid content and a large number of fruits.

### 3.6. Breeding of a New C. chinense Lines Containing High Levels of Capsinoids in the Pericarp

To develop lines with high capsinoid contents, eight $F_5$ lines were selected through repeated individual selection and generational advancement. The capsinoid contents were evaluated for progenies derived from the eight lines, and two individuals derived from SJ 103-4-3-3 and one individual derived from SJ 151-1-4-3 were selected. The capsinoid content in the placentas of the three selected individuals ranged from 6697 µg/gDW to 10,827 µg/gDW, and the content in the pericarp ranged from 2392 µg/gDW to 3532 µg/gDW. The ratio of the pericarp to the placenta capsinoid content ranged from two to five, which means that the pericarp contained as much as 1/2 to 1/5 of the capsinoid content of the placenta.

Comparing the capsinoid contents in the placenta of the three $F_5$ lines with the parental lines, SJ 103-4-3-3-4 (84,325 µg/gDW) and SJ 103-4-3-3-5 (82,197 µg/gDW) showed a significant difference (Figure 5). The capsinoid contents in the pericarp of the three lines SJ 103-4-3-3-4 (14,405 µg/gDW), SJ 103-4-3-3-5 (18,503 µg/gDW), and SJ 151-1-4-3-3 (14,338 µg/gDW) were all significantly higher than that of the parental lines, and the average content per fruit was also significantly different in these lines. The two lines with the highest capsinoid contents, SJ 103-4-3-3-4 and SJ 103-4-3-3-5, were selected.

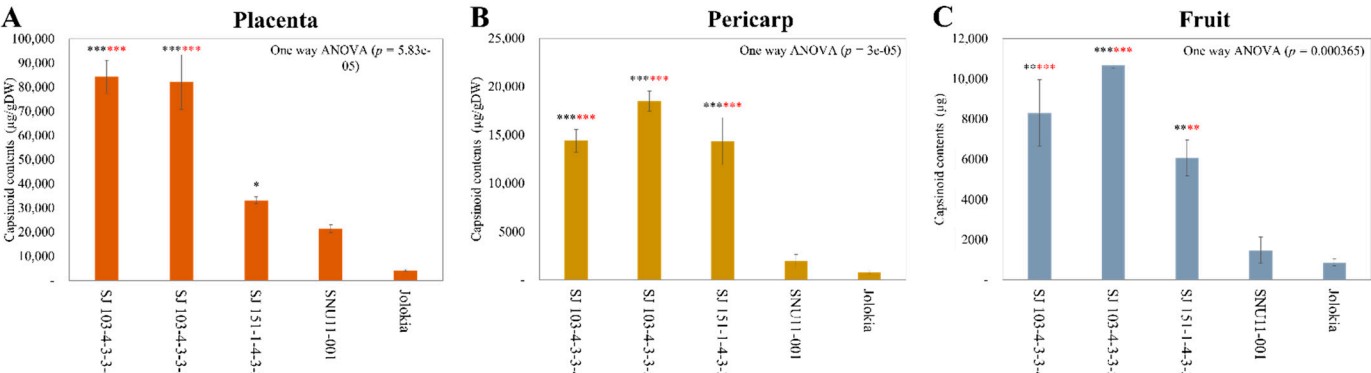

**Figure 5.** Capsinoid contents of the placenta (**A**), pericarp (**B**), and fruit (**C**) in the 'SJ' F7 lines. Bars represent standard error (*n* = 3). Black and red asterisks indicate a statistical difference from 'BJ' and 'SNU11–001' respectively, as revealed using a one–way ANOVA and Fisher LSD, with *p*–values between 0 and 0.001 shown with '***', lower than 0.01, '**' and lower than 0.05, '*'.

## 4. Discussion

This study was conducted to determine whether the gene responsible for the biosynthesis of capsaicinoid in the pericarp of extremely hot peppers can be introduced into other peppers, and whether it can increase the capsinoid contents. We confirmed that lines with very high capsaicinoid contents could be developed using 'BJ', which produces capsaicinoids in the pericarp, and confirmed that this strategy can also be used to develop a variety with a high capsinoid contents.

QTL mapping was performed using interspecific and intraspecific populations to identify the genes that regulate the capsaicinoid contents. Structural genes such as *pAMT*, *C4H*, *4CL*, *CSE*, and *FatA* were previously proposed as candidate genes for this function [16]. In addition, it has been reported that *Erf* (*Ethylene responsive factor*), *Jerf* (*Jasmonic acid and ethylene responsive factor*) [28], *CaMYB31*, *Ankyrin* [9–11,29], and other transcription factor genes are involved in controlling the capsaicinoid contents of the placenta and the pericarp. Until now, the factors regulating the capsinoid content have not been explored; our QTL mapping for capsinoid content and QTL genotype analyses showed that capsinoid content is regulated in a similar manner as the capsaicinoid content.

The average capsaicinoid content per fruit of 'BJ' was higher than that of 'HB', while the content in the placenta of 'HB' was higher than that of 'BJ'. This is likely because capsaicinoids are biosynthesized in both the placenta and the pericarp in 'BJ' but only in the placenta in 'HB'; capsaicinoids are generally produced in the placenta [30], but were also recently reported to be biosynthesized in extremely pungent pericarp [18]. Subsequent studies have reported that capsaicinoid biosynthesis is regulated by genetic factors (QTLs) on chromosomes 4, 6, and 11 [19]. These three QTLs could explain about 50% of the capsaicinoid variation [27]. These QTL–related markers were used to develop lines that produce capsaicinoids and capsinoids in the pericarp. In the 'HJ' population, all individuals were pungent due to the high pungency of the parental lines, but among them, we selected the individual plants harboring the 'BJ' genotypes of the QTL markers to obtain more pungent individuals. QTLs for the pericarp capsaicinoid contents detected in 'HJ' F$_2$ were located between 221.1 Mbp and 245.2 Mbp along chromosome 6, while the QTLs in 'SJ' F$_2$ were 226.8 Mbp to 237.5 Mbp along that chromosome [19]. Here, we showed that the QTL region associated with the capsinoid content in the pericarp was located 226.4 Mbp to 237.5 Mbp along chromosome 6 and considerably overlapped with the QTLs related to the capsaicinoid content in the pericarp (Figure 2).

When we applied the QTL markers identified in the capsaicinoid segregating population to the *pamt* mutant population, the capsinoid contents could be increased (Figure 4). This shows that a genetic factor that increases the capsaicinoid content is also involved in increasing the capsinoid content. QTLs controlling the capsinoid content in the placenta were located 33.4 Mbp to 42.8 Mbp along chromosome 6 in plants containing the wild-type

*pAMT* allele; however, no QTLs were detected on chromosome 6 in the *pamt* mutant segregating group (Figure 2). To explore this relationship in future studies, it will therefore be necessary to perform QTL mapping on a larger *pamt* mutant population.

The three lines selected from the 'HJ' population, HJ 3-1-3-1-2, HJ 3-4-1-3-1, and HJ 3-1-3-1-1, produced significantly more capsaicinoids than 'BJ' (Figure 3). Capsaicinoid contents in both placenta and pericarp of three lines had similar levels with 'HB'. Pericarp contents of three lines did not show statistical differences with 'BJ' but this might be attributable to higher variance in 'BJ' than 'HB'. Even so, the average pericarp capsaicinoid contents of these lines were almost two times higher than those of 'BJ'. In addition, the two lines selected from the 'SJ' population, SJ 103-4-3-3-4 and SJ 103-4-3-3-5, contained significantly more capsinoids in the placenta, pericarp, and fruit than 'SNU11–001' or 'BJ' (Figure 5). It was therefore confirmed that the capsinoid contents can be enhanced by crossing the capsinoid–containing lines with lines containing high levels of capsaicinoids in both the placenta and the pericarp.

Various genetic resources for the production of capsinoids were discovered in domesticated *C. chinense*, *C. annuum*, and *C. frutescens* accessions and used for breeding. 'SNU11–001' contains high capsinoid content in the placenta due to a mutation in the *pAMT* gene [21]. In Japan, *C. annuum* 'CH–19 Sweet' and *C. chinense* 'Aji Dulce' were found to have high capsinoid contents. These lines were crossed with pungent lines containing the wild–type *pAMT* allele, resulting in the development of the capsinoid–containing lines 'HC 3-6-10-11' and 'Maru Salad' [23,25]. Most recently, four varieties of the Dieta series were developed by crossing 'Aji Dulce' with a very pungent *C. chinense* accession [24]. Dieta0011–0301 contained 9.6 mg/gDW of capsinoids and similarly, Dieta0011–0602, Dieta0041–0401, and Dieta0041–0601 contained 15.2 mg/gDW, 19.9 mg/gDW and 15.4 mg/gDW, respectively. This demonstrates that the capsinoid contents can also be increased when an extremely pungent line is used as a parent. Until this study, MAS using the *pamt* mutant allele was the only breeding method for capsinoid cultivar development. The average fruit capsinoid contents of the three lines bred by crossing 'SNU11–001' and 'BJ' were 4.1 times (SJ 151-1-4-3-3; 6066 µg) to 7.5 times (SJ 103-4-3-3-5; 10,666 µg) higher than that of 'SNU11–001' (1467 µg) (Figure 5C). These average contents could be higher than in the parental line because the capsinoid content was also increased in the pericarp, which occupies a larger volume than that of the placenta.

This study demonstrates that a gene regulating capsaicinoid biosynthesis in extremely pungent pepper pericarps can be a useful tool in the breeding of varieties with increased capsaicinoid and capsinoid contents.

**Supplementary Materials:** The following are available online at https://www.mdpi.com/article/10.3390/agriculture11090819/s1, Figure S1: Fruits of parental lines and selected lines, Figure S2: Development of QTL markers to select pungent peppers with high capsaicinoid content in the pericarp, Figure S3: Capsaicinoid contents of the pericarp in 'HJ' $F_2$ and $F_3$ lines, Figure S4: *pAMT* marker for distinguishing the *pamt* mutant from the wild-type allele, Figure S5: Capsinoid content of the pericarp in selected 'SJ' $F_2$ and $F_3$ plants, Table S1: Capsaicinoid contents and QTL genotypes of the 'HJ' $F_2$ and $F_3$ lines, Table S2: Capsaicinoid contents per fruit in 'HJ' $F_5$ lines, Table S3: Capsinoid contents of the placenta, pericarp and fruit in the 'HJ' $F_6$ and $F_7$ lines, Table S4: Capsinoid contents and QTL genotypes of the 'SJ' $F_2$ *pamt* mutant lines, Table S5: Capsinoid contents of the 'SJ' $F_2$ and $F_3$ lines, Table S6: Capsinoid contents of the placenta, pericarp and fruit in the 'SJ' $F_6$ and $F_7$ lines, Table S7: SNP information for the 'HJ' and 'SJ' QTL markers, Table S8: Segregation of *pAMT* alleles in the 'SJ' $F_2$ population, Table S9: KASP mixture components for the *pAMT* genotype analysis, Table S10: KASP PCR conditions for the *pAMT* genotype analysis.

**Author Contributions:** Conceptualization, S.J. and B.-C.K.; data curation, S.J.; formal analysis, S.J.; funding acquisition, J.-H.L. and J.-W.J.; investigation, S.J., M.P. and D.-G.L.; project administration, B.-C.K.; software, S.J.; supervision, B.-C.K.; validation, S.J., M.P. and D.-G.L.; visualization, S.J.; writing—original draft preparation, S.J.; writing—review and editing, B.-C.K. All authors have read and agreed to the published version of the manuscript.

**Funding:** This work was carried out with the support of the "Cooperative Research Program for Agriculture Science and Technology Development (Project No. PJ015881)", Rural Development Administration, Republic of Korea. This work was supported by a National Research Foundation of Korea (NRF) grant funded by the Korean government (MSIT) (No. 2021R1A2C2007472).

**Conflicts of Interest:** The authors declare no conflict of interest.

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
