# Peer review of "Breeding Capsicum chinense Lines with High Levels of Capsaicinoids and Capsinoids in the Fruit"

_agriculture, doi:10.3390/agriculture11090819_

Round 1

Reviewer 1 Report

The manuscript is of great interest but the methodogy is not well clear. The Material and Metods sections are not allignate with the results section !….Please specify all the list of the breeding lines analysed for capsaicinoids and capsinoids …what about the description of the F2 populations? Phenotyping and genotyping results and plots? The fig. 2 seems to be part of Material and methods and the same commenti s related to the F3 population structure to be reported in the M&M section than in results one…..What about experimental design? How do you grown in open field and in greenhouse the plants? The growing techniques and the growing period? the statitical analysis? Al these breeding lines are deposited? Where? Repository?

Al the data are of interest but we need to reduce the confusion reading the article.

Author Response

Dear Editor,

Many thanks for your careful review of our manuscript and for those of the two reviewers. We have revised the manuscript to meet reviewer’s suggestions and feel that the manuscript has been improved by the process.

Below we list the reviewer’s comments in black and the response to those comments in blue. Also, we revised the manuscript and highlighted in blue.

(We attached Word file same as below.)

[Reviewer 1]
The manuscript is of great interest but the methodogy is not well clear. The Material and Metods sections are not allignate with the results section !….Please specify all the list of the breeding lines analysed for capsaicinoids and capsinoids …what about the description of the F2 populations? Phenotyping and genotyping results and plots? The fig. 2 seems to be part of Material and methods and the same commenti s related to the F3 population structure to be reported in the M&M section than in results one…..What about experimental design? How do you grown in open field and in greenhouse the plants? The growing techniques and the growing period? the statitical analysis? Al these breeding lines are deposited? Where? Repository?

Al the data are of interest but we need to reduce the confusion reading the article.

  • Specify all the list of the breeding lines analyzed for capsaicinoids and capsinoids.
    : We accepted the reviewer’s suggestion and added the capsaicinoids and capsinoids contents of all breeding lines in Table S1 to Table S6. Figure S3 and Figure S5 show the capsaicinoid contents of ‘HJ’ breeding lines and capsinoid contents of ‘SJ’ breeding lines, respectively. However, two graph charts was not enough to specify phenotype.
  • Describe the phenotype, genotype, and plots of the ‘SJ’ and ‘HJ’ F2
    : Table S8 contains pAMT genotype analysis of ‘SJ’ F2 population and Table S4 shows capsinoids content and QTL genotype result of ‘SJ’ F2 pAMT mutant plants. We added Table S1 to describe the phenotype of ‘HJ’ F2 breeding lines.
  • Figure 2 seems to be part of material and methods and the same comments related to the F3 population structure to be reported in the M&M section than in results one.
    : We accepted the reviewer’s suggestion. Figure 2 was rearranged to Figure 1, and we described details of the pedigree method in ‘Materials and methods- 2.1 Pedigree method for development of ‘SJ’ and ‘HJ’ lines’ instead of ‘2.1 Plant materials’. In addition, F3 generation of ‘SJ’ and ‘HJ’ was also added in this section.
  • How do you grow in open field and in greenhouse the plants? The growing techniques and the growing period?
    : We added ‘2.2 Growing region and period’ in materials and methods section. Growing condition of glasshouse and greenhouse was mid-tech condition, and open field was low-tech condition. Most of the periods was from spring to autumn, but only F5 was grown in Thailand from autumn to next spring.
  • Describe the statistical analysis.
    : We used one-way ANOVA to check the capsaicinoid and capsinoid contents were higher than parental line significantly. We added the sentences in m&m.
  • All these breeding lines are deposited? Where? Repository?
    : Seeds of all breeding lines are stored in Seoul National University.

[Reviewer 2]

The manuscript brings important information on the genes regulating capsaicinoid biosynthesis in Capsicum and the successful breeding of new genotypes containing high level of capsaicinoid and capsinoids in their fruits. Overall, the manuscript is well-written. Introduction summarizes resent research related to the topic. The aim is well stated. The experiment was well designed and executed. The approach to methodology used seem appropriate. The results are well discussed. I have only minor comments to the authors on how to improve the work.

  • Abbreviations
    - (CAPS marker Line 111 should be Cleaved Amplified Polymorphic Sequences (CAPS))
    - Line141, ‘HPLC’ should be High Performance Liquid Chromatography (HPLC)
    : We accepted the reviewer’s suggestions and changed the words.
  • Materials and methods should contain information of the performed statistical analyzes and the equipment used (model, city, country).
    : HPLC was performed by Ultimate3000 HPLC (Thermo Dionex, USA) at National Instrumentation Center for Environmental Management (Seoul, Republic of Korea).
  • Lines185-185 about QTLs mapping made by Park et al .2019, should be moved from Results to Discussion. Reference should be converted to the number.
    : We accepted the reviewer’s suggestions and that moved to discussion, and changed reference style.
  • Figure 1. Full name of ‘SJ’ should be given in the photo caption.
    : Figure 1 was replaced to Figure 2. Figure 1 had full name of ‘SJ’.
  • The writing style of each article listed in References should be checked according to the requested style by the journal.
    : Sorry for the mistakes. We changed some articles of reference style.
  • Supplementary Figure 1. Please explain the bars in the photos.
    : We added the information of white bars, 2 cm.

Reviewer 2 Report

The manuscript brings important information on the genes regulating capsaicinoid biosynthesis in Capsicum and the successful breeding of new genotypes containing high level of capsaicinoid and capsinoids in their fruits. Overall, the manuscript is well-written. Introduction summarizes resent research related to the topic. The aim is well stated. The experiment was well designed and executed. The approach to methodology used seem appropriate. The results are well discussed. I have only minor comments to the authors on how to improve the work.

Abbreviations used in the text for the first time should be given their full name, e.g. (CAPS marker Line 111 should be Cleaved Amplified Polymorphic Sequences (CAPS)).

Materials and methods should contain information of the performed statistical analyzes and  the equipment used (model, city, country).

Lines185-185 about QTLs mapping made by Park et al .2019, should be moved from Results to Discussion. Reference should be converted to the number.

Line141, ‘HPLC’ should be High Performance Liquid Chromatography (HPLC).

Figure 1. Full name of ‘SJ’ should be given in the photo caption. 

The writing style of each article listed in References should be checked according to the requested style by the journal.

Supplementary Figure 1. Please explain the bars in the photos.

Author Response

Dear Editor,

Many thanks for your careful review of our manuscript and for those of the two reviewers. We have revised the manuscript to meet reviewer’s suggestions and feel that the manuscript has been improved by the process.

Below we list the reviewer’s comments in black and the response to those comments in blue. Also, we revised the manuscript and highlighted in blue.

We attached Word file same as below.

[Reviewer 1]
The manuscript is of great interest but the methodogy is not well clear. The Material and Metods sections are not allignate with the results section !….Please specify all the list of the breeding lines analysed for capsaicinoids and capsinoids …what about the description of the F2 populations? Phenotyping and genotyping results and plots? The fig. 2 seems to be part of Material and methods and the same commenti s related to the F3 population structure to be reported in the M&M section than in results one…..What about experimental design? How do you grown in open field and in greenhouse the plants? The growing techniques and the growing period? the statitical analysis? Al these breeding lines are deposited? Where? Repository?

Al the data are of interest but we need to reduce the confusion reading the article.

  • Specify all the list of the breeding lines analyzed for capsaicinoids and capsinoids.
    : We accepted the reviewer’s suggestion and added the capsaicinoids and capsinoids contents of all breeding lines in Table S1 to Table S6. Figure S3 and Figure S5 show the capsaicinoid contents of ‘HJ’ breeding lines and capsinoid contents of ‘SJ’ breeding lines, respectively. However, two graph charts was not enough to specify phenotype.
  • Describe the phenotype, genotype, and plots of the ‘SJ’ and ‘HJ’ F2
    : Table S8 contains pAMT genotype analysis of ‘SJ’ F2 population and Table S4 shows capsinoids content and QTL genotype result of ‘SJ’ F2 pAMT mutant plants. We added Table S1 to describe the phenotype of ‘HJ’ F2 breeding lines.
  • Figure 2 seems to be part of material and methods and the same comments related to the F3 population structure to be reported in the M&M section than in results one.
    : We accepted the reviewer’s suggestion. Figure 2 was rearranged to Figure 1, and we described details of the pedigree method in ‘Materials and methods- 2.1 Pedigree method for development of ‘SJ’ and ‘HJ’ lines’ instead of ‘2.1 Plant materials’. In addition, F3 generation of ‘SJ’ and ‘HJ’ was also added in this section.
  • How do you grow in open field and in greenhouse the plants? The growing techniques and the growing period?
    : We added ‘2.2 Growing region and period’ in materials and methods section. Growing condition of glasshouse and greenhouse was mid-tech condition, and open field was low-tech condition. Most of the periods was from spring to autumn, but only F5 was grown in Thailand from autumn to next spring.
  • Describe the statistical analysis.
    : We used one-way ANOVA to check the capsaicinoid and capsinoid contents were higher than parental line significantly. We added the sentences in m&m.
  • All these breeding lines are deposited? Where? Repository?
    : Seeds of all breeding lines are stored in Seoul National University.

[Reviewer 2]

The manuscript brings important information on the genes regulating capsaicinoid biosynthesis in Capsicum and the successful breeding of new genotypes containing high level of capsaicinoid and capsinoids in their fruits. Overall, the manuscript is well-written. Introduction summarizes resent research related to the topic. The aim is well stated. The experiment was well designed and executed. The approach to methodology used seem appropriate. The results are well discussed. I have only minor comments to the authors on how to improve the work.

  • Abbreviations
    - (CAPS marker Line 111 should be Cleaved Amplified Polymorphic Sequences (CAPS))
    - Line141, ‘HPLC’ should be High Performance Liquid Chromatography (HPLC)
    : We accepted the reviewer’s suggestions and changed the words.
  • Materials and methods should contain information of the performed statistical analyzes and the equipment used (model, city, country).
    : HPLC was performed by Ultimate3000 HPLC (Thermo Dionex, USA) at National Instrumentation Center for Environmental Management (Seoul, Republic of Korea).
  • Lines185-185 about QTLs mapping made by Park et al .2019, should be moved from Results to Discussion. Reference should be converted to the number.
    : We accepted the reviewer’s suggestions and that moved to discussion, and changed reference style.
  • Figure 1. Full name of ‘SJ’ should be given in the photo caption.
    : Figure 1 was replaced to Figure 2. Figure 1 had full name of ‘SJ’.
  • The writing style of each article listed in References should be checked according to the requested style by the journal.
    : Sorry for the mistakes. We changed some articles of reference style.
  • Supplementary Figure 1. Please explain the bars in the photos.
    : We added the information of white bars, 2 cm.

Round 2

Reviewer 1 Report

The manuscript was revised on the basis of the comments. It is now ready to be published.